# “They Just Need to Come Down a Little Bit to Your Level”: A Qualitative Study of Parents’ Views and Experiences of Early Life Interventions to Promote Healthy Growth and Associated Behaviours

**DOI:** 10.3390/ijerph17103605

**Published:** 2020-05-21

**Authors:** Marita Hennessy, Molly Byrne, Rachel Laws, Caroline Heary

**Affiliations:** 1Health Behaviour Change Research Group, School of Psychology, National University of Ireland Galway, H91 TK33 Galway, Ireland; molly.byrne@nuigalway.ie; 2Institute for Physical Activity and Nutrition (IPAN), School of Exercise and Nutrition Sciences, Deakin University, Geelong, Victoria VIC 3220, Australia; r.laws@deakin.edu.au; 3School of Psychology, National University of Ireland Galway, H91 TK33 Galway, Ireland; caroline.heary@nuigalway.ie

**Keywords:** childhood obesity, prevention, parent, qualitative, interview, thematic analysis, infant feeding, intervention, pregnancy, infancy

## Abstract

The first 1000 days is a critical window of opportunity to promote healthy growth and associated behaviours. Health professionals can play an important role, in part due to the large number of routine contacts they have with parents. There is an absence of research on the views of parents towards obesity prevention and the range of associated behaviours during this time period. This study aimed to elicit parents’ views on early life interventions to promote healthy growth/prevent childhood obesity, particularly those delivered by health professionals. Semi-structured interviews were conducted with 29 parents (24 mothers, 5 fathers) who were resident in Ireland and had at least one child aged under 30 months. Data were analysed using reflexive thematic analysis. Two central themes were generated: (1) navigating the uncertainty, stress, worries, and challenges of parenting whilst under scrutiny and (2) accessing support in the broader system. Parents would welcome support during this critical time period; particularly around feeding. Such support, however, needs to be practical, realistic, evidence-based, timely, accessible, multi-level, non-judgemental, and from trusted sources, including both health professionals and peers. Interventions to promote healthy growth and related behaviours need to be developed and implemented in a way that supports parents and their views and circumstances.

## 1. Introduction

The first 1000 days—the period from conception to a child’s second birthday—is a critical window of opportunity to promote the health of women and their children [1,2]. Early life environmental conditions set the foundations for health states in later childhood and adulthood [3]. Eating and physical (in)activity-related habits, behaviours, and patterns are established during the first two years of a child’s life [2,4,5], which track into later childhood and young adulthood [6,7,8,9]. Parents play a key role in the healthy growth and development of their children, especially during the early years when children are dependent on them for a range of needs including nutrition and care [10,11].

Early life factors—such as high maternal pre-pregnancy weight, smoking during pregnancy, high infant birth weight and rapid weight gain, infant feeding mode, and the early introduction of solid foods—are associated with the development of childhood overweight/obesity [12,13,14] and are amenable to intervention during this period. Interventions that can be embedded into ongoing practice and existing systems are needed [15]. How to integrate obesity prevention into existing service structures was identified as the fifth most important priority for childhood obesity prevention research [16].

Parents have regular contact with health professionals during pregnancy and the first two years of life as part of routine health visits [15]. Early life childhood obesity prevention interventions delivered by health professionals show some evidence of effectiveness, particularly in relation to behavioural outcomes [17]. The acceptability of such interventions by parents and health professionals is poorly reported [18]. Research is required to understand how to support and engage parents, and how to support practitioners to support parents [16].

To date, research has examined parents’ views of interventions around discrete behaviours or topics, such as infant feeding and weight management during pregnancy [19,20,21,22,23,24,25,26], often focusing on the views of primiparous mothers. Limited attention has been paid to parents’ overall perceptions of interventions to promote healthy growth, as well as specific topics such as active play/physical activity and sedentary behaviour. Much existing research has tended to focus on pre-school aged children and older [27,28]. That said, one study has, however, explored parental views of obesity risk identification during infancy, finding that while parents find it acceptable; they also experience high levels of parental responsibility, fear of judgement, and self-blame [29,30].

This study aims to address these current gaps in the literature by eliciting parents’ views of early life interventions to promote healthy growth and/or prevent obesity and those delivered by health professionals in particular. We defined intervention broadly as any advice, help, or support regarding a child’s weight and/or growth, and associated behaviours, from any source, whether solicited or unsolicited. Findings will aid the development and/or adaptation of early life interventions to promote healthy growth.

## 2. Materials and Methods

### 2.1. Design and Study Setting

This study formed part of a larger study, which also sought to explore parents’ perceptions of healthy growth and associated behaviours during the first 1000 days. The findings of the broader study will be reported elsewhere. Face-to-face, semi-structured interviews were conducted. This methodology was best suited to address the study aim. The philosophical underpinnings of this research combined constructivist ontology with interpretivist epistemology, with the aim of providing insight into participants’ constructions of reality and the ways in which such constructions are socially and culturally situated [31]. Ethical approval was granted by the NUI Galway Research Ethics Committee (Ref: 17-May-08). The Standards for Reporting Qualitative Research were followed in writing this paper [32] (See Appendix A).

A Parent Advisory Group (three mothers, one father), convened to advise the research team for the purposes of this study, reviewed and refined the study protocol and study materials (available on OSF [33]). They were recruited through a local community-based parent and toddler group.

This study was conducted in Ireland, where over 60,000 babies are born annually [34]. Women ordinarily resident in Ireland are entitled to free maternity care [35]; care is also available privately, and postnatal care is provided to all women. There are 25 routine contacts between parents and the health service between conception and a child’s second birthday. See Appendix A for further context.

### 2.2. Participants and Recruitment

We included parents and/or primary caregivers of children aged under 30 months living in Ireland. Parents were purposively recruited, on a range of criteria, to ensure variety in the sample: mothers/fathers, socio-economic backgrounds, prima/multi-parous parents, parents whose babies were breast-/formula-fed, location–urban/rural, ethnic/cultural backgrounds, and body mass index (BMI). This information was captured in a brief demographic questionnaire administered to participants before each interview. Participants were asked to self-report their height and weight, and BMI was calculated.

Participants were recruited through community groups (e.g., Mother and Toddler Groups, groups that target fathers/men) and social media. Gatekeepers in community groups provided a copy of the information sheet and consent form to potential participants. Parents were asked to make contact with the lead researcher if interested in participating or if they required more information. Upon agreeing to take part, participants completed a consent form, and a suitable time and location for the interview was agreed.

### 2.3. Conduct of Interviews

Interviews were conducted by MH, a 38-year-old female, experienced in public health and health services research, who was neither a parent/ever pregnant. An interview guide (Appendix A) included questions on: views of childhood obesity and perceived importance of the issue and healthy infant growth, understanding of behaviours associated with healthy growth, experiences around infant growth and associated behaviours, views about interventions to prevent childhood obesity/promote healthy growth—particularly those delivered by health professionals. Participants, if multi-parous, were asked to respond to questions thinking about their youngest child (i.e., the child aged under 30 months) but also to reflect on their experiences with other children. The interview guide was piloted with a mother and father, following review by the Parent Advisory Group; no amendments were necessary, and both pilot interviews were included in the analysis. Interviews lasted between 45 min and 2.5 h (average = 86 min). They were digitally recorded, and following each interview, participants were provided with a document outlining sources of information and support and a copy of the signed consent form. The interviewer also made detailed field notes.

We deemed that sufficient sampling occurred when the major categories showed depth and variation in terms of their development. We focused on achieving a high level of “information power” (i.e., the more information the sample holds, the lower the number of participants needed) [36] rather than “saturation”, which is incompatible with reflexive thematic analysis [37].

### 2.4. Analysis

Interview audio recordings were transcribed verbatim, checked for accuracy and de-identified. Transcripts were imported into NVivo (QSR International Pty Ltd. Version 11, 2015) and data analysed using reflexive thematic analysis [31,38]. MH read and re-read and then coded each transcript (coding material specially addressing the research aim of this particular paper) and subsequently examined the codes and collated data to identify potential themes in discussion with CH, MB, and RL—all parents of young children. MH then checked and refined the candidate themes against the dataset before developing a detailed analysis of each theme and deciding on an informative name for each. We interviewed mothers and fathers purposively, to get as complete a picture as possible of views and experiences of parents and thereby enhance the rigour and credibility of the findings. We also engaged in peer debriefing as a study team, maintained an audit trail through comprehensive notes, and provide thick description of the context and findings.

## 3. Results

We interviewed 24 mothers and 5 fathers, with infants/children ranging in age from 7 weeks to 25 months, with two of the interviewees also being pregnant at the time of interview. One mother and one father were co-habiting; the remainder were from separate families. Four of the participants were known, to varying extents, to individual members of the study team; however, the lead researcher did not have a relationship with any of them, beyond acquaintance. Participant characteristics are outlined in Table 1.

Two central themes, with several sub-themes, were generated from the data: (1) Navigating the uncertainty, stress, worries, and challenges of parenting under scrutiny and (2) Accessing support in the broader system (Table 2).

### 3.1. Theme 1: Navigating the Uncertainty, Stress, Worries, and Challenges of Parenting under Scrutiny

This theme describes the challenges parents experienced in navigating this stage of life. New parents felt vulnerable, full of uncertainty, and experienced stress and worry as a result. This was compounded by both felt and perceived judgement, stigma, and guilt in relation to their parenting. Most spoke about the challenges of parenting and trying to maintain balance between caring for their child(ren), work and other commitments. All of this influenced their need for support, and their ability to access information and support, but also to put the acquired knowledge and/or skills into practice. With growing experience, parents’ confidence increased.

#### 3.1.1. Finding Your Way around in the Dark

Almost all participants spoke about how parents experienced various stresses relating to their parenting role. Some highlighted how mothers are particularly vulnerable in the early postnatal period. Many mentioned their lack of knowledge as a first-time parent in general, but also particularly in relation to breastfeeding and introducing solids. Most felt it was important to know what to expect and be prepared when it came to breastfeeding, with some highlighting the importance of knowing what to expect and be prepared in relation to child weight and general parenting. Some stated that they had felt well prepared antenatally: *“…one of the best things in that group was the whole understanding and expectation that you may not find it easy or any way doable to breastfeed for the first two or three or four days”* (D04). Others, however, felt that they did not know what to expect, one saying *“it’s like you’re finding your way around in the dark”* (D01).

While many felt that breastfeeding was often perceived as “easy”, the reality, initially at least, was quite different and stressful. Various challenges were experienced: getting breastfeeding established, physiological problems, and knowing that their baby was getting enough milk and gaining sufficient weight. Many described how they persevered with breastfeeding, or with other aspects of child feeding (e.g., introducing solids/baby led weaning) to a lesser extent. For the majority, when they saw the outcomes of their efforts, it gave them confidence in their ability/decision. Many spoke of the pride they felt when their babies were doing well as a result of their efforts. For some, however, particularly in instances where a baby’s weight needed to increase, they had to let go of their desire to exclusively breastfeed.

Most participants had concerns about their child’s weight/growth at one point or another, particularly insufficient weight gain. Some spoke about the stress associated with weight and/or developmental checks, especially if their child had an issue identified, and questioned their parenting ability: *“…I was just making sure, trying to make sure that we were feeding her enough and that she would have enough. So I was a bit anxious about the midwives’ visit to weigh her”* (M13). A few spoke about becoming unnecessarily “obsessed” about weighing their children. Other concerns centred on infant’s sleep, fear around giving birth, and generally “keeping them alive”.

Most parents highlighted the importance of early intervention and parental support: *“If you can give them the support and information early on, in their pregnancy, and that you start building those layers of confidence…”* (M04), with some noting that parents with less resources may require additional support.

#### 3.1.2. Felt and Enacted Judgement, Stigma, and Guilt

Judgement, or fear of being judged by others, was experienced by almost all participants—especially in relation to infant feeding issues, and parenting more generally. This impacted on their views and experiences, particularly in terms of who they discussed issues with/sought advice and support from. Many negative encounters with health professionals, peers, and family members were relayed.

Participants spoke about maternal feelings of guilt or shame around infant feeding/parenting. Discussions often centred on the breastfeeding versus formula-feeding “divide”, and feelings of guilt or shame if women could not meet their breastfeeding goals, with decisions to breastfeed for an extended period/follow baby-led weaning less frequently mentioned. Some spoke about negative relationships with, or support from, their own mothers or other women, particularly if these women had not breastfed/met their own breastfeeding goals. Many felt that:


*You breastfeed and you get like flack for it, or you’re defending yourself or you’re covering yourself up. Or… you don’t breastfeed, and you’re giving a bottle to your child… there’s just so much guilt and shame and judgement around so many different things with parenting.*
(M16).

Participants also mentioned comments from others around child size/weight and felt that parents were often, or could be, judged by others if their child had a weight issue.

Some participants spoke about judgement experienced—directly or indirectly—in online forums, particularly in relation to infant feeding. Some were cautious of online forums, and just “lurked”, and while the pros outweighed the cons for some, others stopped using them altogether. The importance of a supportive, open, and inclusive environment within in-person groups was also mentioned by some participants. This included breastfeeding support groups that permitted those who did not exclusively breastfeed to participate: *“Some people were expressing and some people were doing half formula half breastfeeding. And that just led to much more, being able to talk about what’s hard about it. And that actually makes you breastfeed longer”* (M12).

Some parents felt that their parenting skills would be judged by others if they sought support because *“when you’re a first-time mum, you put a lot of pressure on yourself…You feel like you can’t ask for help because then everyone will think you’re useless”* (M17). Many spoke about instances where they did not disclose or tell the truth about certain issues *“for an easy life”* (M01) or fear of judgement: *“you think, they’re going to get me locked up. They’re going to take my baby away”* (M04). This could be in relation to infant feeding (practices/experiences, or not following advice to top-up with formula) primarily, or infant sleep and activity during pregnancy, for example.

Some felt that how breastfeeding is promoted/discussed led to judgement, stigma, shame, or guilt. Some participants spoke of interactions with those who had strict or “militant” views about breastfeeding: *“…it’s great that there’s movement towards trying to encourage it, but (.) some people take that to the extreme”* (M03). There was a general feeling that *“they [haven’t] really cracked it yet, how to talk about breastfeeding”* (M15). Participants noted that while *“Breastfeeding is promoted…they’re not doing anything to support it”* (M13), citing societal, cultural, and economic reasons that shaped attitudes and norms around infant feeding and child growth, and undermined women’s beliefs in their ability to feed their babies. The need for more support for breastfeeding was stressed by many—to make it *“more normal”* (M16) and to provide more practical supports to mothers/parents, including around enabling them to make informed infant feeding decisions. Participants viewed exposure to breastfeeding and positive support received as enablers.

The majority stressed the importance of non-judgemental, positive, and reassuring information and support, and that a *“wider cultural shift [is needed]….I think we need to be much more supportive of parents in general”* (M02). This was primarily in relation to breastfeeding (but also child growth/development, child feeding and sleep), and from health professionals and peers/friends. They valued the confidence boost it provided, often by confirming that the issues they were experiencing were “normal” and/or that they were doing a good job.

#### 3.1.3. Increasing Confidence and Learning to Trust Your Instincts

Most parents relayed how their confidence grew with support and time/experience. Some spoke specifically about how their confidence increased as they grew into their parenting role: *“We’re a lot more confident in everything we do with [Niall], you know what I mean, compare to poor little [Rosie] who was like our first and we stressed about everything”* (M17). Many highlighted how learning what was “normal” for babies (e.g., in relation to their size, feeding, sleep), while acknowledging that all babies are different (even within families), and finding or accepting their own normal, was important. Many spoke about the need for parents to trust their own judgement, and “go with their guts” or instincts, sometimes citing that they do not, but over time, they learn to trust them more. A few parents spoke about not asking questions, particularly as a first-time parent, and how they would ask if they were in similar situations again.

Some participants spoke about being “less wound up” on subsequent pregnancies—not worrying about things as much (or having the time to) and/or doing as much information seeking. Some highlighted that increased knowledge/confidence meant that they did not need or rely on such support on subsequent pregnancies, some mentioned that health professionals held less power, while others stated that people just “leave you at it”:


*…for first time mums there’s so much interfering and advice giving and shoving it down your throat. And it’s just such an anxious time already because you don’t know what to do… Whereas the second time people go oh yeah look she knows what she’s doing and they leave you at it, and EVERYTHING is easier.*
(M02)

### 3.2. Theme 2: Accessing Support in the Broader System

Theme 2 describes how parents navigated and accessed supports in the broader system. They had a range of sources, with health professionals (during routine contacts and antenatal classes primarily), peers/family, online, partners/husbands, and “own research” most often mentioned. Health professionals encountered included general practitioners (GPs), midwives, public health nurses (PHNs), lactation consultants, and, to a lesser extent, obstetricians, pediatricians, and practice nurses. Parents primarily recalled discussions with health professionals around infant/child feeding, as well as child growth assessments, with tummy time also frequently mentioned; other topics received less attention and often only took place when parents brought them up. Further details are provided in Appendix A.

Parents highlighted structural, inter-personal, and personal barriers and facilitators to accessing support. Health professional support was often perceived difficult to access. Parents appreciated the accessibility of online peer support. Levels of trust varied in information sources. Parents were often in receipt of lots of advice from different sources, which was often unhelpful if not clear and tailored to their needs. Relationships and relatability were highly valued. Resultantly, parents valued a combination of both professional and peer support.

#### 3.2.1. Ability to Access and Engage with Supports/Services

Most participants spoke about the importance of being able to access and/or engage with services and supports, especially when you needed them. There were mixed reports about the difficulty (or ease) in getting appointments with health professionals. Lack of access to lactation consultants in hospital was a significant issue: *“one lactation consultant who does the whole of [Maternity Hospital 3], which is not enough”* (M07). Difficulties accessing other breastfeeding supports, lack of choice in what health professionals (e.g., PHN) you get to see, and delays in between PHN checks or falling through the system were also highlighted. Many felt that services and health professionals (especially midwives, GPs, and PHNs) were “stretched”, which impacted on availability of health professionals, as well as time available to discuss issues and provide support, primarily around breastfeeding. Some women reported feeling *“on the clock”* during appointments and that interactions lacked meaningful engagement: *“… at every appointment they have a little blurb that they have to say that they explained X Y and Z to you, like breastfeeding’s best for baby…… They tick their box and they hand you the leaflet…”* (M01). Similar experiences were reported in relation to developmental checks, which were often merely seen as *“a tick-box exercise to see is your child progressing against certain milestones”*. (M08)

Many participants spoke about having to access supports privately, particularly for lactation consultants, tongue tie releases, and/or, to a lesser extent, antenatal classes. The majority had accessed such private supports but noted the cost implications (though some could claim back from private health insurer) and how this was not possible for everyone.

Being able to contact PHNs, lactation consultants, and peer breastfeeding supports by phone (call/text) with any queries was mentioned positively by several participants. Some also spoke about the benefits of the internet for its 24/7 support, particularly in relation to breastfeeding. The ability to have opportunistic discussions with PHNs at peer support groups was valued by some.

Many participants spoke about doing their own research to supplement (and sometimes check) information from other sources and/or to find alternatives that would work for them: *“…they’ll give you some [information] and then I feel like it was up to me to go off and find out more”* (M17). Many also spoke about the importance of being able to ask questions, particularly in relation to being able to engage with a health professional, ask them questions, or challenge their advice. The need and ability to judge the quality of information received was also stressed.

#### 3.2.2. Relationships and Relatability are Key

Many highlighted the importance of continuity of care and support, primarily in relation to health professionals; this facilitated trust and relationship building, which was not the experience for many. Those on homebirth, DOMINO (Domiciliary Care in and Out of Hospital), or Early Transfer Home schemes (see explanations in Appendix A) generally reported positive experiences in this regard, as did those who had extra contact with health services, e.g., due to gestational diabetes: *“She [community midwife] knew me. She knew the history with me and the birth, everything. I wasn’t talking to another random stranger”* (M09). A few participants spoke about the importance of contacts made during the antenatal period (e.g., during classes), which extended postnatally and *“it meant that I had a contact for when I was postnatal and things weren’t going particularly well”* (M11).

Participants had mixed levels of trust in health professionals, within and across disciplines. While some noted that as a first-time parent you have total faith in health professionals and “go with what they say”, this trust sometimes waned. This could be due to negative experiences such as feeling unsupported during encounters and/or receiving (perceived) incorrect or conflicting advice, including inaction or mis- or late diagnoses (e.g., in relation to tongue-tie and allergies/intolerances). This often left parents seeing that *“they are just people with just information, and they don’t necessarily have all the answers”* (M02). Some participants spoke about trusting peers and/or friends or family, sometimes more than health professionals: *“they get where you’re coming from so then you tend to believe them more”* (M18), or a perception that health professionals *“can deal with the normal stuff, but they can’t deal with anything outside the realm of normal”* (M02). Others reported trusting health professionals more than information received through online forums. The majority stressed the need for a mix of information sources.

Social support (or lack thereof), encompassing emotional, informational, and practical support from peers and/or family members, was discussed by all participants. Face-to-face peer support groups (as well as online, e.g., for breastfeeding and baby led weaning) were generally highly regarded, as you could *“…see how other people are getting on and how are you getting on… you’re all in the same boat”* (M09). Peer support via WhatsApp, text or phone call was also mentioned and valued.

Many participants (mothers and fathers) stressed the importance of dads as a source of practical and emotional support for mothers, particularly around breastfeeding. Some noted that fathers might not be as supportive when difficulties were encountered (instead placing emphasis on the needs/health of the mother) or when babies were breastfed for an extended period. Some highlighted the need to and/or value of engaging dads and also the role they play in their children’s lives. Despite this, three of the participant fathers spoke about feeling undermined/patronised by health professionals: *“they treat you like you’re a bit thick you know. And you’re kind of, no I’m actually mad into this”* (D02).

Most parents stressed the value of support from someone who understands, with experience, and who could empathise, as *“they’ve had kids themselves, so they understand, you know, that not everything is going to happen by the book”* (M03). Many spoke about reassurance received from health professionals, primarily around their child’s growth at checks but also in relation to gestational weight gain or child feeding. Some spoke about the importance of health professionals listening to, and being respectful of, parents. A few participants spoke about the need for health professionals to see the “bigger picture” (e.g., patterns of growth, rather than once-off assessments) and also respect parents views and wishes, particularly in relation to developmental checks/growth assessment. Some felt that PHNs took a narrow view and needed to *“come down a little bit to your level”* (M16), whereas GPs took more account of context: *“[GP] had no issue [with the child’s growth] because he knew more of the background I guess of the family”.* (M22). Some also felt that midwives provided more holistic care to both mother and baby compared to other health professionals, and *“… always really human and just asking well how are YOU doing”* (M12).

#### 3.2.3. Everyone Has an Opinion

Almost all participants described instances where they were on the receiving end of unhelpful attitudes, advice, and support or pressure to engage in certain practices, from a wide range of sources and whether sought or not. Again, much of this centred around infant feeding and breastfeeding in particular. Participants highlighted the value of clear communication from health professionals and the importance of consistent messaging/advice between different health professionals and/or other information and support sources.

Conflicting, unclear, or inconsistent advice from health professionals was often reported. This was primarily in relation to infant/child feeding, for example, *“…they were trying to encourage her [sister-in-law] to get to six months [before introducing solids], but they were telling me it was okay from four months…”* (M01). The internet was also a place where *“you’d see conflicting things. Often it’s people’s opinions”* (M03).

Parents voiced concern that while health professionals promoted breastfeeding, they were quick to offer formula. Pressure to give a breastfed baby formula was frequently mentioned, especially in relation to the early postnatal period in hospital, with formula top-ups arising from concerns about weight loss, as well as perceived widespread availability and unprompted provision of formula:


*hospitals need to be less pushy with their formula……And I didn’t get any support on that either you know as in like there was no expressing equipment or anything like that, so it was just you know a case of do you want a bottle, do you want a bottle…And then even when we were leaving the hospital they gave us a box of bottles.*
(M10).

Health professionals were seen to prioritise weight over breastfeeding:


*…one of the most stressful things is those four or five weeks after you give birth, where the first, Jesus like if your baby doesn’t make their weight back up in the first ten days, God help you…*
(M18)

Lack of support from family members around breastfeeding was also raised, as was unsolicited advice from members of the public: *“… the hardest thing that I’ve found about being a parent, is everybody else”* (M07).

Some participants felt that they received no new, useful, or relevant information, during contacts with health professionals or via online/printed resources. More tailored approaches were suggested: *“general information probably doesn’t work so well with me. It’s like people actually figuring out what’s YOUR specific issue and solving that”* (M11). Many participants spoke about lack of knowledge or skills amongst all categories of health professionals, particularly concerning breastfeeding and child growth patterns. Many spoke about the need for health professionals, particularly GPs and PHNs, to be more pro-active in providing support and/or raising issues—even if it could be hard for parents to hear, particularly in relation to a mother’s weight and related behaviours, or a child’s weight/growth. Some, however, felt that if there was an issue, their health professional would raise it (particularly around weight). Some felt that health professionals required more education/training to be able to raise and/or discuss issues and keep up-to-date, particularly around infant feeding and especially baby led weaning.

#### 3.2.4. Information and Support that Meets Parents Where They Are at

Most parents stressed the need for information to be practical and realistic, with some mentioning that they did not follow advice “to the letter” if this was not the case. They also welcomed being shown how to do things. Practical support in relation to breastfeeding, particularly in hospital to get it established or from lactation consultants, was highly valued. Such demonstration of behaviours was also appreciated in relation to hands-on activities in antenatal classes and weaning workshops or during routine checks (by health professionals demonstrating behaviours, and/or providing resources/leaflets which did). Videos were also popular, especially amongst fathers. Participants also liked guidance to be clear, not too directive, and that they could adapt to their own situation. Some spoke about how some antenatal classes engaged fathers really well, with some also noting how they and/or their partners were highly engaged when information around the science of breastfeeding was presented.

Most participants highlighted the value of timely information and support. They also expressed a preference for a staged approach to information provision, rather than receiving lots in one go: this involved receiving information specific to a child’s developmental stage/needs (at the right time). Some also highlighted that information only becomes relevant when going through a particular issue/stage, e.g., information provided at antenatal classes only becomes relevant when the baby arrives.

Some participants also made explicit reference to the mode of delivery of information or support, with some preferring face-to-face as they could ask questions. As mentioned earlier, group support was valued by many (in-person and online), while a few preferred one-to-one support, as either they did not like being in groups and/or appreciated the tailoring to their individual circumstances.

## 4. Discussion

This study set out to elicit parents’ views on early life interventions to promote healthy growth/prevent childhood obesity, particularly those involving health professionals. Two central themes were generated from the data. The first relates to parents themselves “Navigating the uncertainty, stress, worries, and challenges of parenting whilst under scrutiny”. Parents, particularly first-time parents, felt vulnerable during this period, in which there is a lot of uncertainty, stress, and worry, often exacerbated by both felt and perceived judgement, stigma, and guilt in relation to their parenting/feeding practices, which were often under scrutiny. With the right support and time, however, their confidence increased, underscoring the importance of supporting and reassuring parents during this key life stage. The second theme “Accessing support in the broader system” demonstrates that while parents are receptive to, and would welcome, such support, particularly around feeding, several critical components need to be considered.

The first theme reinforces much of what is known about the transition to parenthood, a challenging time for both mothers and fathers [39,40,41,42,43,44]. Parents often felt they had a lack of knowledge and felt insufficiently prepared for the realities of parenting, particularly breastfeeding, which has been noted elsewhere [45,46,47]. Consistent with previous research, our study findings highlight the judgement, stigma, and guilt/shame that parents perceive around parenting and infant growth [29,48], but particularly in relation to infant feeding [23,44,49,50,51,52,53,54,55]. Judgement within online forums was also noted, similar to recent studies concerning breastfeeding [56,57], child growth [58], and mothering [59]. Parents require non-judgemental support and reassurance to allay concerns and build their infant feeding/parenting self-efficacy.

Many participants in our study, however, noted how health professionals often prioritised weight gain over all else, with infant formula often provided in hospitals if infants were not gaining sufficient weight. This was also observed in a study involving women with a higher weight in Sweden [60]. Women could also become unnecessarily “obsessed” with having their babies weighed, for example at baby clinics/groups provide, even though it provided opportunities for them to feel pride in their maternal achievement and “good mothering, as previously observed in the literature [61,62,63]. It is critical that interventions are framed and developed and/or adapted and implemented in ways that consider these factors and provide non-judgemental support to parents. Interventions that promote healthy growth in young children in a way that is appropriate to their stage of development, focuses on key behaviours, and positively supports parents are needed.

Participants, including those who exclusively breastfed and those who did not, highlighted how better approaches to breastfeeding promotion were needed, as some women could feel pressured to breastfeed [64,65] and feel judged for not doing so [66]. Having to justify reasons for breastfeeding can create divisions between formula-feeding and breastfeeding mothers and can impact on social support for all [67]. Increasingly, there are calls in the wider literature for a change in how we communicate about breastfeeding to enhance rates: a move away from the perception that “breastfeeding is best”, emphasis on wider values other than the health benefits of breastfeeding, and a message that every feed matters [68]. Including unbiased, evidence-based support around formula-feeding, where appropriate, may be important to reduce the risk of alienating women and improve reach and retention of interventions [54,69]. It is also important that the social context is taken into consideration, given that judgement can also come from health professionals, peers, family members, and the general public. Our findings also demonstrate that over time, with support and/or with subsequent children, parents become more confident.

The second theme generated in the study focuses on what parents require in terms of information and support. Support around infant feeding, particularly breastfeeding, was highlighted in particular. This is perhaps unsurprising, given that most participants interviewed (or their partners) had breastfed their babies, coupled with the fact that Ireland has one of the lowest rates of breastfeeding globally [70], with initiation rates of 60% and 49% of “any” and “exclusive” breastfeeding, respectively [34]. Introducing solids and infant growth were other areas where parents felt they needed more support. Information and support received from health professionals during pregnancy was highly varied; as noted elsewhere, e.g., in relation to weight and/or gestational weight gain diet/nutrition and physical activity [71,72,73,74]. Resultantly, parents expressed a desire for health professionals to be more pro-active in raising issues, with many feeling interactions to be tick-box exercises, something that health professionals themselves acknowledge [75].

Barriers highlighted by parents in this study such as lack of time, knowledge, and resources may account for this and should be addressed within interventions. Indeed, health professionals themselves have identified lack of knowledge, confidence and/or relationships with women, time, continuity of care, and resources as barriers to them supporting women around a range of health behaviours [75,76,77,78,79,80]. Similarly, maternal and child health nurses in Australia raised concern about parental receptiveness to discussions about child weight in routine practice [81]. This could also be a potential reason why participants felt that interactions were often tick-box exercises. In line with previous research, participants were generally supportive of health professionals raising issues such as maternal weight and pregnancy weight gain [25,26,74,82,83] and to child growth/weight [29], even if they would be hard to hear, if done in a non-judgemental, supportive way. Continuity of care, and the associated development of trusting, respectful relationships and rapport with health professionals, were highly valued by parents in this study, similar to prior research concerning antenatal care [74], weight and/or gestational weight gain [84,85,86], and infant feeding [47,64]. Thus, parents desire more support and for health professionals—with whom they have good rapport with—to be more pro-active in raising such issues.

It is clear from our study that there is no one intervention and/or approach that will work for all parents. Parents wanted high quality information, in a variety of formats (e.g., face-to-face/online; individual/group). Consistent with other research, particularly around infant feeding, parents felt that they were often in the receiving end of conflicting and confusing advice [20,87,88] and were critical of the quality of information received, often supplementing with other sources. Research by Lupton has also found that while parents value online support, they highly value expert advice and expressed the desire for increased information and support offered by healthcare professionals [89]. Some parents in our study valued one-one-one support, whereas others valued group interaction; this has been noted in previous breastfeeding research [90,91]. Women value practical demonstrations and being shown how to feed their baby (particularly time patiently spent watching them feed their baby), rather than be told how to feed them [64]. Indeed, some studies also note how parents value a combination of professional and peer support in relation to breastfeeding [62,92,93]. Our study further highlights that it is important to support all parents, including fathers, who can often feel excluded/patronised during encounters with health professionals [42,46,94,95,96] but particularly those who are first-time parents. It is crucial that inadequacies in universal provision are considered, such as greater access to breastfeeding support and information during routine contacts with parents. Efforts should be made to improve delivery of interventions that promote healthy growth and associated behaviours during routine care. Support, however, must be practical, realistic, tailored, evidence-based, timely, accessible, and from trusted sources, including both health professionals and peers. This study builds the evidence base concerning parents’ views and experiences of early life interventions to promote healthy growth, and the range of associated behaviours, including active play/physical activity and sedentary behaviour. Specifically, it provides insight into parents’ experiences of interventions concerning the range of these behaviours in the context of child growth, particularly those delivered by health professionals/within routine care, which has received limited attention to date.

## 5. Strengths and Limitations

The strengths of this study include: the inclusion of both mothers and fathers, primiparous and multi-parous parents; the examination of a broad range of factors related to healthy growth in young children, not simply infant feeding; and the involvement of parents in the design and conduct of the study. Several limitations should be noted, however. Despite our best efforts to engage a diverse group of parents, our sample had a high number of parents with higher education levels and those who breastfed/were breastfeeding their children. Furthermore, many of the participants worked in health and social care roles so this may have influenced their views. In addition, while we used various strategies to engage fathers, including relationship development and targeted communications, we encountered challenges in recruitment. This is not uncommon within studies relating to the promotion of healthy growth, where fathers are often under-represented [97,98]. While only five participants were fathers, however, the data generated from those interviews was rich. The role of social desirability bias was considered at the outset of this study, particularly given the “surveillance” and judgement parents can experience. That said, every effort was made to establish rapport with parents during the interviews and ensure that they knew that we were interested in hearing their views and opinions. While we were interested in views of interventions, particularly those involving health professionals, we only interviewed parents. Further studies should examine health professionals’ views and also wider perspectives, including those of grandparents/other caregivers.

## 6. Conclusions

The first 1000 days of life is a critical window of opportunity to support the development of healthy growth and associated behaviours in children. Interventions involving health professionals in particular can be important, given the large number of routine contacts between parents and the health service during this period. This study elicited parents’ views and experiences of interventions, in general and specifically those delivered by health professionals. Parents reported a range of sources of information and support, and mixed experiences with each. They were receptive to, and would welcome, support during this critical time period, particularly around feeding. Support, however, needs to be practical, realistic, evidence-based, timely, accessible, non-judgemental, and from trusted sources, including both health professionals and peers. Various levels of support and intervention are required, at individual/intra-personal, inter-personal, organisational, community, and policy levels. Interventions to promote healthy growth and related behaviours need to be developed and implemented in a way that supports parents and their views and circumstances.

## Figures and Tables

**Table 1 ijerph-17-03605-t001:** Participant characteristics.

**Relationship to Child**	Mother (n = 24), Father (n = 5)
**Age range**	<25y (n = 1), 25–30 (n = 2), 31–39 (n = 21), 40+ (n = 5)
**Marital status**	Married (n = 23), co-habiting (6)
**Parity**	1st child (n = 14), 2nd/subsequent child (n = 15)
**Age of infants ***	7 weeks to 25 months
**Feeding mode ***	All breastfed, to varying extents (1 h–24+ months); 25 exclusively breastfed (8w–6 months; average = 5 months, with some babies <6 months still being breastfed)
**Parent BMI range**	<18.5 (n = 1), 18.5–24.9 (n = 15), 25.0–29.9 (n = 12), ≥30 (n = 1)
**Parent born in Ireland**	Yes (n = 23), no (n = 6)
**Ethnicity**	White Irish (n = 25), other (n = 4)
**Highest level of education**	Secondary school (n = 3), post-secondary technical qualification (n = 2), degree (n = 6), postgraduate (n = 18)
**Employment status**	Currently on maternity leave (n = 11), employee (n = 14), self-employed/unemployed/other (n = 4)

* Most recent child, if parent of more than one child.

**Table 2 ijerph-17-03605-t002:** Overview of themes and sub-themes.

Theme	Sub-Theme
1. Navigating the uncertainty, stress, worries, and challenges of parenting under scrutiny	Finding your way around in the dark
	Felt and enacted judgement, stigma, and guilt
	Increasing confidence and learning to trust your instincts
2. Accessing support in the broader system	Ability to access and engage with supports/services
	Relationships and relatability are key
	Everyone has an opinion
	Information and support that meets parents where they are at

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
