# Peer review of "“They Just Need to Come Down a Little Bit to Your Level”: A Qualitative Study of Parents’ Views and Experiences of Early Life Interventions to Promote Healthy Growth and Associated Behaviours"

_ijerph, 2020, doi:10.3390/ijerph17103605_

Round 1

Reviewer 1 Report

Reviewer #1:

  1. Thank you for the opportunity to review this manuscript. Overall, this is an interesting study examining parents’ views of early life interventions to promote healthy growth and/or prevent obesity, particularly those delivered by health professionals. I have some suggestions and comments below for the authors to consider:

General suggestions:

  1. Page 1, line 12: add a space after the word “Correspondence:”.
  2. Page 4, line 150: to be consistent replace D1 to D01.
  3. Do not use words abbreviation in the description of the study, such as don’t – line 229; “wasn’t” – line 288; didn’t and wasn’t – line 376.
  4. Page 7, line 283: Please review the sentence “Particularly in relation to [be] able to engage…”

 I provide minor comments below for the authors' consideration:

Introduction:

  1. Page 2, lines 48-49: “Parents have regular contact with health professionals…” – Please specify the country (ies).

Materials and methods:

  1. Page 2, lines 76-77: “A Parent Advisory Group (three mothers, one father)…” – How this was defined?
  2. Page 2, line 88: Was the demographic questionnaire used previously validated in Ireland population?
  3. Page 3, lines 105-106: “Interviews lasted between 45 minutes and 2.5 hours” – Please explain this large differences in the duration of the interviews.

Results:

  1. Page 3, line 125: If the sample was purposely recruited, could you please explain why only five parents were interviewed?

 Discussion:

  1. The discussion is well written and discusses the findings nicely in light of existing evidence. The implications in terms of targets for future interventions are also discussed.

Strengths and limitations:

  1. Even though both, mothers and fathers participated in the study, there was a small number of fathers participating – Please mention that participation rates among fathers relative to mothers is usually low in scientific studies.

Reviewer 2 Report

Review of “They just Need to Come Down a Little bit to your Level”: A Qualitative Study of Parents’ Views and Experiences of Early Life Interventions to Promote Healthy Growth and Associated Behaviours

This paper covers an extremely important and sensitive topic that is potentially key to childhood obesity. Parents play a key role to their child’s nutrition education during these early stages, from infant feeding (ideally exclusive breast feeding) to complementary feeding after the age of 6 months, as the authors refer to briefly in their introduction. Have the authors gathered such information? It would be interesting to see the views and perspectives of parents that adopt recommended regimes compared to those that don’t.

Introduction:

Line 33: 1000 months covers the first 34 months of life, hence >2 years of age and <34 years. The authors however refer to the child’s 2nd birthday. Please correct accordingly.

Lines 43-44: please add “are” before amenable

Line 54: What do the authors mean by “discrete behaviors”? Some examples should be given.  

Lines 62-67: the aim of the study should be more specific and measurable. The broad definition and from any source, may mean that non-professional advice is included (even information derived from any internet sites?)? these may enter bias in the study and nutrition educators caution on their use. Were these separately analyzed? Meaning parents that gathered information from any means compared to those that were informed by health care professionals?

Methods:

Line 71: “constructivist ontology with interpretivist epistemology”… this is a very specific term that requires further explanation with references in order for the readers to be informed. Where has this method been used before and how is it used?

Lines 76-77: more information on the background of the parents that formed this advisory group should be provided. How were they selected? What were the requirements? Why was this group developed…

Why were only 24 mothers and 5 fathers interviewed? It seems that the sample size is quite limited for generalizable conclusions. For women with multiple parity (2 children, n this study) the questions were focused on the second child I assume. This must be explained in methods section.  

Does Ireland have baby friendly hospitals? Are IYCF guidelines followed and if so how? Which variables do health care professionals focus? This question is based on the fact that women perceive BF easy, although they realize it is not. It is a usual misconception that proper guidance from trained professionals help resolve. Please expand.

Overall many important points are derived, which are areas that ned to accounted for when addressing parental and child needs when designing educational programs, including public health care facilities. These, however, should be somehow categorized (ie by parity and educational level…) since these are variables that may affect various factors examined. Many parents have different views and beliefs based on their own background, this should be therefore accounted for, despite the small number of individuals included.

In the discussion, many points are covered with many being extremely controversial (ie formula feeding and BF). Tis should be clearly defined by the authors.

Table: 1 hour of trying to BF is actually no BF.

Reviewer 3 Report

1, Need to provide more information about the representativeness of participants.

2, According to guide of the semi-structured interviews, they collected rich infromation in the interview. But it seems they did not report it clearly in the manuscript.

3,Need more accurate information, need tables to report the distribution of participants and clearly state it in the manuscript. For example, line 142 - 146 in the manuscript. "Almost all participants spoke about how parents experienced various stresses relating to their parenting role. Some highlighted how mothers are particularly vulnerable in the early postnatal period. Many mentioned their lack of knowledge as a first-time parent, but also particularly on breastfeeding and introducing solids. Most felt it was important to know what to expect and be  prepared when it came to breastfeeding, and, to a lesser extent, child weight and general parenting."

4,  Need to analysis the association between characteristics of participants and the two central themes (or even the sub-themes).

Reviewer 4 Report

Dear Ms Ms. Pearl Yang:

Thank you for inviting me to review the manuscript titled “They just Need to Come Down a Little bit to your Level”: A Qualitative Study of Parents’ Views and Experiences of Early Life Interventions to Promote Healthy Growth and Associated Behaviours”, after reading the manuscript, my comments are below:

  • A theoretical and conceptual construction of the rationale and needs to conduct research on exploring “parents’ views of early 62 life interventions to promote healthy growth or prevent obesity and those delivered by health 63 professionals in particular”.
  • The authors need to explicate clearly any sampling strategies they used to select parental interviewers systematically, that helps to preclude biases.
  • In analyzing the interviewing scripts, any guiding research procedures that the authors had employed to help enhance objectivity in achieving the “themes” they obtained as genuinely reflecting the reality of parents’ perceptions.
  • In Discussion, the authors need to apply what they have found in application to the domain of public health.

Best

The reviewer

Round 2

Reviewer 1 Report

Dear,

Thank you for the opportunity to review this manuscript.

The authors have answered my questions and I have no further comments and suggestions to provide.

Author Response

Thank you once again for taking the time to review our paper.

Reviewer 2 Report

Thank you for your revision and point by point discussion. 

The paper presents important information that health care providers should consider. 

Author Response

Thank you once again for reviewing our paper.

Reviewer 3 Report

The authors have tried to address the comments.

Author Response

(The authors gave the same response as above.)
